# Upper Esophageal Sphincter Dysfunction in Children with Type 1 Laryngeal Cleft after Failed Primary Cleft Repair

**DOI:** 10.3390/biom14010015

**Published:** 2023-12-21

**Authors:** Corey Baker, Casey Silvernale, Christopher Hartnick, Claire Zar-Kessler

**Affiliations:** 1Pediatric Gastroenterology, Hepatology and Nutrition, Connecticut Children, Hartford, CT 06106, USA; 2Pediatric Gastroenterology, Hepatology and Nutrition, Mass General for Children, Boston, MA 02114, USAczarkessler@mgh.harvard.edu (C.Z.-K.); 3Massachusetts Eye and Ear Infirmary, Boston, MA 02114, USA

**Keywords:** laryngeal cleft, upper esophageal sphincter (UES), high-resolution esophageal manometry, oropharyngeal dysphagia

## Abstract

Changes in pharyngeal and upper-esophageal-sphincter (UES) motor dynamics contribute to swallowing dysfunction. Children with type 1 laryngeal clefts can present with swallowing dysfunction and associated symptoms which may persist even after the initial endoscopic intervention. This study sought to characterize pharyngeal and esophageal motor function in children with type 1 laryngeal clefts who had persistent presenting symptoms after their initial therapeutic intervention. We retrospectively analyzed high-resolution esophageal manometry studies of children ≤ 18 years old with type 1 laryngeal clefts who had an esophageal manometry study performed for persistent symptoms after an initial repair. A total of 16 children were found to have significantly increased UES resting pressure, UES pre- and post-swallow maximum pressures, and duration of UES contraction during swallows in comparison to nine age-matched controls of children without pharyngeal anatomical abnormalities. There was no difference between UES residual pressures or pharyngeal dynamics between the two groups. UES resting and residual pressures did not correlate with VFFS in penetration and aspiration scores of children with type 1 laryngeal clefts status post repair. Our study is the first to identify specific changes in UES motor function in patients with type 1 laryngeal cleft post initial repair.

## 1. Introduction

A laryngo-tracheo-esophageal cleft, or laryngeal cleft, is a rare congenital malformation characterized by an abnormal, posterior, sagittal communication between the larynx and the pharynx, at times extending downward between the trachea and the esophagus. The estimated annual incidence of a laryngeal cleft is 1/10,000 to 1/20,000 live births, accounting for 0.2% to 1.5% of congenital malformations of the larynx [1]. The Benjamin and Inglis classification of cleft severity is the most well recognized classification [2], designating type 1 clefts as a supraglottic interarytenoid cleft that spares the cricoid cartilage progressing to type 4 clefts which include a laryngo-esophageal connection extending into the thoracic trachea and as far as to the carina [3]. The involved anatomical structures along with cleft severity contribute to symptom severity [4]. Patients with laryngeal clefts can have a wide variety of symptoms including stridor, hoarse cry, swallowing difficulties, aspirations, cough, dyspnea and cyanosis, and respiratory distress [1]. Scoring systems grading the ability of pediatric patients without laryngeal clefts to protect their airway have been established, most notably the Penetration-Aspiration (PA) Scale via results of a barium swallow study [5] whose use has been described in children with laryngeal clefts before and after primary repairs [6]. 

Treatment options of clefts mainly depend on their grade or severity. The majority of type 1–2 laryngeal clefts have been shown to respond to medical management such as thickening of feeds [7]. However, a minority do not respond to medical management, and like larger clefts, require endoscopic or surgical interventions [7,8]. Endoscopic interventions are variable and include the placement of sutures, CO_2_ laser repair, or injecting a filling substance (such as Pro into the cleft such as [8,9]). Despite these interventions, a minority of patients with small clefts experience persistent symptoms post primary endoscopic repair. Their persistent symptomology highlights the fact that these symptoms are multifactorial at baseline and that there could be other possible contributing factors, including upper-esophageal-sphincter (UES) dysfunction. 

The upper esophageal sphincter is an area of high pressure located in the upper digestive tract that forms a proximal barrier of the gastrointestinal tracts from the pharynx. This high-pressure zone is made up of three muscles, the cervical esophagus, the cricopharyngeus, and the inferior pharyngeal constrictor, or thyropharyngeus. The UES is considered a high-pressure zone which in concordance with the pharynx appropriately relaxes allowing the passage of food into the esophagus for further digestion [10]. Dysfunction of these muscles results in the incomplete relaxation and/or early closure of the upper esophageal sphincter, resulting in proximal dilatation and the trapping of contents leading to secondary problems of feeding difficulties with or without penetration and/or aspiration [11]. 

There are multiple investigatory techniques used to evaluate oropharyngeal dysphagia such as electromyography, a fiberoptic endoscopic evaluation of swallowing (FEES), and a videoflouroscopic swallow study (VFSS), with the latter being most commonly used [12,13]. Esophageal manometry is an investigatory tool that has been established in its use of evaluating for esophageal motility disorders by providing real-time high-resolution imaging of pressurization and relaxation of the upper esophageal sphincter, esophageal body, and lower esophageal sphincter [14]. Evaluation and interpretation of pharyngeal manometric patterns using high-resolution solid-state pharyngeal or esophageal catheters have characterized normal swallowing in healthy adult volunteers [15], identified specific changes in UES pressures in adults with dysphagia [16,17], and characterized changes in pharyngeal and upper-esophageal-sphincter function in response to speech therapy interventions in particular adult patient populations [18]. More recently, Damrongmanee et al. [19] reviewed high-resolution esophageal manometry of 142 children who had normal swallows on VFSS. Their findings suggested that the motor dynamics of swallowing may mature with age since UES resting pressures were higher than previously reported UES resting pressures in neonates and lower than those of adults. Additionally, children had similar UES nadir pressure to that of adults but higher pharyngeal pressures. Children with abnormal swallowing on VFSS had significantly higher UES nadir pressures compared to those children with normal swallowing.

Children with laryngeal clefts can present with a variety of clinical symptoms prior to therapeutic interventions with a subset of patients having persistent symptoms despite such invasive therapies having taken place. Given the recent emergence of the use of high-resolution esophageal manometry in the evaluation of pediatric oropharyngeal dysphagia, we sought to characterize pharyngeal and UES motor dynamics in children with type 1 laryngeal clefts after primary endoscopic repair who had persistent symptoms by comparing them to similarly aged children without a history of a laryngeal cleft or other pharyngeal anatomic abnormalities. We hope to identify differences in high-resolution esophageal manometry in this specific patient population in comparison to the age-matched control children. Characterizing the manometric variances in children with type 1 laryngeal clefts with persistent symptoms after their initial repair may allow clinicians to better understand the underlying pathophysiology of these patients’ symptoms, provide increased efficacy of therapeutic interventions, and thus potentially improving clinical outcomes. Additionally, our findings may lead to identifying new targets for future research in enhancing diagnostic algorithms or in the development of novel therapeutic interventions. 

## 2. Materials and Methods

### 2.1. Identification of Patients

This is a retrospective chart review using an RPDR query tool for patients under 18 years old with laryngeal cleft type 1 status post repair with persistent symptoms, and who were evaluated with esophageal manometry at our tertiary care center from 2015 to 2019. Patients with laryngeal clefts were diagnosed with type 1 clefts with laryngoscopy performed by a single pediatric otolaryngologist, and which was subsequently repaired via endoscopic injection laryngoplasty, or approximation via endoscopic sutures performed by the same provider. Patients’ sex, comorbidities, age of diagnosis, type of cleft repair, persistent symptoms prior to manometry, time of esophageal manometry after repair, and manometry findings were collected. Age-matched controls who had manometry performed for clinical indications without esophageal anatomical abnormalities were identified via chart review over the same time period. The upper esophageal sphincters of nine age-matched controls were analyzed after six other patients were excluded due to associated diagnoses of congenital esophageal abnormalities including congenital esophageal stenosis or achalasia. 

### 2.2. High-Resolution Esophageal Manometry

All reviewed esophageal manometry studies used the Medtronic ManoScan ESO High-Resolution Manometry System software v3.3, (Medtronic, Minneapolis, MN, USA) which generates real-time esophageal topography pressure plots of swallows. All patients of both study groups had a Medtronic high-resolution small-diameter esophageal catheter placed transnasally while under general anesthesia upon the completion of an esophagogastroduodenoscopy (EGD). This solid-state esophageal catheter is 2.75 mm in diameter and has 36 circumferential pressure sensors that are spaced 7.5 mm apart from one another. To ensure adequate placement of the catheter with its distal tip terminating in the stomach, prior to the completion of the EGD, the endoscopist estimated the location of the esophagogastric junction by visually identifying the squamocolumnar junction or Z-line. The distal tip of the endoscope was placed adjacent to the squamocolumnar junction, and using the distance markers on the outside surface of the endoscope, the distance at the lower lip of the patient was measured. The manometry catheter was then transnasally advanced 1–5 cm past this measured distance at the lower lip. The endoscope was then reinserted to the oral cavity and re-intubated the upper esophagus to visually confirm that the catheter was appropriately placed within the esophagus. The patient was then taken to the recovery room where the manometry study was started once the medical team deemed the patient alert enough to be able to safely swallow liquid and communicate developmentally appropriately, which was most often 45–60 min from the patient’s arrival at the recovery unit. All manometry studies were performed per protocol by the MGHfC Neurogastroenterology team which includes physicians, nurses, and a child-life specialist. Upon initiation of the manometric study, adequate placement of the catheter is confirmed or adjusted if necessary through visual identification of the diaphragm and subsequent positioning of the catheter’s distal sensor 1–3 cm below it. Once each patient was in their most quiescent state, a 20 s resting period was captured where no swallows or the most minimal amount of swallows occurred. If tolerated, each patient was given 5 mL of water by mouth via a syringe to swallow once on command. A total of at least 10 swallows were captured for analysis. Given the young age of the majority of the patients in both study groups, the ability for each patient to swallow on command or tolerate liquids given by syringe, was limited. Therefore, in order to capture any swallowing data, patients were offered to swallow liquid through their preferred vessel (bottle, sippy cup, etc.). In an effort to minimize preceding multiple swallows, the medical team and patients’ family members made every effort to limit the patient to one swallow at a time. 

### 2.3. Upper-Esophageal-Sphincter Analysis

Resting and residual UES pressures were obtained via the patient’s esophageal manometry procedural report in the electronic record. The techniques applied to identify and analyze pharyngeal and UES parameters were conducted using previously described techniques by Kammer et al. via high-resolution esophageal manometry [18]. Particular parameters of the UES that were analyzed pre-swallow included maximum pressure of velopharynx, mesopharynx, the UES pressure at rest, and the preopening maximum UES pressure or pre UES pressure. During relaxation, parameters measured were the minimum UES pressure and UES residual pressure, which is the highest UES pressure during maximum mesopharyngeal pressure. Additionally, the duration between the onset of pressure in the velopharynx until maximum post-closure pressure of the UES (duration of swallow), the onset of UES relaxation to its closure (UES relaxation), and the time difference between the swallow duration and UES relaxation or time the UES is not relaxed during the swallow were measured. The maximum pressure of the UES after its relaxation or post UES pressure was also measured [18]. Each swallow of each patient was individually analyzed and averaged making up a value of for one patient (*n* of 1). Since this is a retrospective review and the protocol for esophageal manometry does not necessitate full visualization of the pharynx and/or UES, not all of the swallows could be incorporated in this UES analysis. Therefore, for each swallow, the entire UES and/or individual upper-esophageal-sphincter/pharyngeal parameters were excluded from the analysis if they were unable to be appropriately evaluated due to absent or inadequate visualization of these parameters for each individual swallow for each patient. This analysis was performed by a single research-team member.

### 2.4. Video Fluoroscopic Swallow Study and Penetration-Aspiration Scale Scores

Reports of VFSS studies of laryngeal cleft patients were selected to analyze possible associations with UES pressures. The VFSS selected for each patient was performed after the primary laryngeal cleft repair. It was also the closest VFSS study that the patient had to his or her esophageal manometry study. There were no UES interventions during the time between the initial repair and the esophageal manometry. Each VFSS was performed and reported as per the standard protocol by speech pathologists and radiologists. This includes that patients were to be provided with thin liquid or nectar thick liquid diets per VFSS protocol. Using the VFSS report generated by our speech pathology colleagues, a score for the penetration aspiration (PA) scale was assigned for each patient. The PA scale is a standardized scoring system measuring the degree of penetration or aspiration of ingested materials into the respiratory tract during a barium swallow study. The scale ranges from a score of 1 to 8 with each number correlating to particular findings on VFSS and to the patient’s reaction, or lack thereof, to an aspiration event. A score of 1 indicates that no swallowed contrast entered the airway during the VFSS. A score of 2 or 3 indicates that the contrast entered the airway but remained above the vocal cords. A PA score of 2 was given when the contrast was ejected from the respiratory tract while a score of 3 indicates that it was not ejected. A score of 4 or 5 indicates that the contrast entered the airway and makes contact with the vocal cords. A PA score of 4 correlates with the contrast being ejected while a score of 5 was given when the contrast was not ejected from the airway. PA scores of 6–8 are considered to be aspiration events since the contrast enters the airway and crosses below the vocal cords. A PA score of 6 was given when the contrast was ejected from the respiratory tract; a score of 7 indicates that the contrast was not ejected, while a PA score of 8 indicated that the contrast was not ejected and that there was no response from the patient [5]. Individual PA scores for each patient with type 1 laryngeal clefts after primary repair were then plotted in association with their respective individual mean UES residual and resting pressure. PA scores were then grouped into three categories: normal or no penetration or aspiration of ingested material (a PA score of 1), penetration (a PA score of 2–5), and aspiration (a PA score of 6–8). Statistical analysis of the differences of the mean UES residual and resting pressures between these three groups were then performed. 

### 2.5. Statistical Analysis 

UES analysis was compared between the patients with laryngeal clefts and the age-matched controls. VFSS and esophageal manometry of laryngeal cleft patients were compared to each other to assess correlation of UES pressures and degree of clinical symptoms. Values are represented as mean ± SEM (standard error of mean). Statistical analysis was performed using Prism 7 (GraphPad software, Inc., La Jolla, CA, USA). Statistical significance was assessed using Student’s *t*-test. *p*-values < 0.05 were regarded as significant. Linear regression was used to assess association of pharyngeal and UES dynamics to degree of aspiration via the patient’s PA score in laryngeal cleft patients post primary repair. 

## 3. Results

### 3.1. Patient Characteristics 

After the RPDR search, two patients diagnosed with congenital esophageal abnormalities including esophageal stenosis and trachea–esophageal fistulas were excluded from analysis resulting in sixteen patients who were identified as having high-resolution esophageal manometry performed due to persistent symptoms after primary repair of the laryngeal cleft. All 16 patients were initially diagnosed with Type 1 laryngeal clefts. Patients were diagnosed with laryngeal clefts on average at 26.75 months, 43.25% were female, and the average time of esophageal manometry after primary endoscopic repair was 13.25 months. The majority of patients were repaired via endoscopic suture (75%) with the remainder having an endoscopic injection (25%). The presenting diagnoses or symptoms reported necessitating further evaluation via esophageal manometry were aspiration events (68.75%), followed by dysphagia (18.75%), reflux (6.25%), and cough (6.25%). Additional diagnoses or symptoms that were not reported as the presenting diagnosis or symptom included reflux (62.5%), asthma (37.5%), airway pathologies (31.25%), constipation (18.75%), heart murmur (12.5%), food allergies (6.25%), and hypotonia (6.25%). Comparing patients with type 1 laryngeal clefts post repair to age-matched controls, there was no major difference in genders between both groups. There was no significant difference in age at the time when esophageal manometry was performed with age-matched controls (*n* = 9) having this procedure performed on average at age 36.3 months compared to patients with a laryngeal cleft post repair (*n* = 16) presenting at 44.9 months (*p* = 0.3784). The most common presenting symptom associated with performing an esophageal manometry in both groups was aspiration; although, the age-matched control group had a larger proportion of patients with dysphagia (45%) compared to those in the laryngeal cleft group (18.75%) (Table 1). 

### 3.2. UES Manometry Characteristics 

UES residual and resting pressures were not compared to age-dependent standardized pressures, since these currently do not exist in the pediatric population. The residual UES pressures showed no difference between the age-matched controls (control) and laryngeal cleft group (LC). On the other hand, resting UES pressures were elevated in patients with laryngeal clefts post repair (*n* = 16) compared to normal (*n* = 9), while there were no differences in UES residual pressures (Figure 1). Further UES analysis (Table 2) showed no difference in maximum pressures in the velopharynx (LC: 135.5 mmHg (*n* = 10), control: 109.4 mmHg (*n* = 5), *p* = 0.2409), mesopharynx (LC: 142 mmHg (*n* = 12), control: 112.2 mmHg (*n* = 8), *p* = 0.1132), or in minimum UES pressures (LC: 2.1 mmHg (*n* = 12), control: −1.3 mmHg (*n* = 8), *p* = 0.2223) between the two groups. Pre UES maximum pressure was notably elevated in the laryngeal cleft group (91.2 mmHg) compared to the control group (66.7 mmHg, *p* = 0.0308), as well as post UES pressure (179.5 mmHg compared to 134.8 mmHg, *p* = 0.0118). The duration of swallows along with the amount of time the UES was relaxed during the swallowing were not statistically different between both groups (LC: 1.03 s (*n* = 12), control: 1.02 s (*n* = 7), *p* = 0.8716; (LC: 0.68 s (*n* = 12), control: 0.79 s (*n* = 7), *p* = 0.1014), respectively. However, the time that the UES was not relaxed during the swallow was significantly increased in the laryngeal cleft group (0.35 s) compared to the control group (0.23 s, *p* = 0.0031). 

### 3.3. Association of UES Pressures with Penetration and Aspiration Scores 

Increasing penetration- and aspiration-scale scores obtained via VFSS were categorized into groups. The three groups of normal (PA score of 1), penetration (PA scores of 2–5), and aspiration (PA scores of 6–8) were not found to be statistically different from one another in correlation with mean resting and residual pressures of the UES in the laryngeal cleft patients, (Figure 2). 

## 4. Discussion

Our study is the first to characterize pharyngeal and UES motor dynamics in pediatric patients with a type 1 laryngeal cleft who were persistently symptomatic after their initial endoscopic repair. When compared to age-matched controls, children who remain symptomatic after their first laryngeal cleft repair had a higher maximum resting UES pressure, a higher UES pressure before and after its relaxation during swallowing, as well as a longer duration of UES contraction during swallows. 

Distinct changes in UES dynamics in patients with particular diagnoses are continuing to be more appreciated. As previously mentioned, the 2021 study by Damrongmanee et al. [19] showed changes in UES integrated relaxation pressure (IRP) and UES nadir pressure in children with abnormal VFSS. Additionally, a hypertensive UES has been reported in patients with Trisomy 21 [20]. Abnormalities in UES characteristics and function is also reported in more common diagnoses where a hypertensive UES was described in adult patients with GERD [21], and less frequent UES contractile reflex response was found in patients with supraesophageal reflux disease [22]. In 2015, Chavez et al. suggested that 57% of adult achalasia patients had been found to have some type of UES abnormality (most often having a hypertensive UES (50%)), compared to patients without achalasia (42%, *p* = 0.04) [23]. 

Our finding of an elevated UES pressure in children with type 1 laryngeal clefts after primary repair may be inherent in this diagnosis as the trachea and esophagus both originate from the foregut [24]. Perhaps the hypertensive upper esophageal sphincter and increased contraction period is part of an underlying neuromuscular or discoordination which was reported to occur in 38% of patients with laryngeal clefts [25]. Another possibility is that our findings are due to the laryngeal cleft repair itself. While there are varying severities of laryngeal clefts, our studied group only included type 1 clefts that do not extend past the true vocal cords [2]. The upper esophageal sphincter is slightly distal to the true vocal cords suggesting that even the most severe type 1 cleft would not be adjacent to the upper esophageal sphincter. Therefore, the manipulation of the larynx for a type 1 cleft repair would not involve the upper esophageal sphincter [9,26] and is unlikely to have a direct effect on its tonicity or functional capacity. 

Interventions to repair clefts are considered to be overall effective but have variable outcomes. While there is no standardization on measurements of success for post-operative outcomes [27], penetration and aspiration scores have frequently been used to assess swallowing improvement [28]. Interestingly, our noted differences in the UES of laryngeal cleft patients after their repair did not correlate with the severity in penetration aspiration scores. This may suggest that our UES findings may not be solely responsible for the abnormal VFSS findings. Interestingly, this is different from the previously mentioned 2021 Damrongmanee et al. [19] study which showed a correlation in differences in UES motor dynamics in children with abnormal VFSS findings. However, the 2016 study by Ferris et al. [29] which evaluated clinical-symptom severity in children with UES motor dynamics and imaging produced findings that did not show significant correlation in UES abnormalities with penetration and aspiration scores on VFSS, but did with symptoms of oropharyngeal dysphagia. These variable outcomes highlight the need for standardization in how these investigatory techniques are used and interpreted in the pediatric population. 

While our study is the first to identify novel findings in UES pressure characteristics in children with type 1 laryngeal clefts, there are a few limitations. First, this was a retrospective study that was analyzed by one researcher who was not blinded to the patients. Therefore, selection and interpreter biases may exist. However, the cleft diagnosis, repair, esophageal manometry, and VFSS interpretations were completed by other clinicians not involved in the study analysis. Additionally, it could be suggested that the use of anesthesia to place the esophageal manometry catheter may alter the results given the known side effects of anesthesia on smooth muscle. However, we believe this did not affect our results as we waited an extended period of time after anesthesia until the patient was alert, and both comparison groups were exposed to anesthesia in a similar fashion. Similarly, an additional limitation was the variability, and most often inability, of the majority of patients to effectively swallow once on command, which is ideal for the most accurate esophageal manometry study. However, this is an inherent issue with performing esophageal manometry studies in this age group due to their normal maturation level. But again, this uncontrollable limitation is unlikely to affect our results since both groups were of similar age and thus had an evenly distributed amount of limitation in regard to the patient’s ability to follow commands. 

In conclusion, this is the first study to characterize the upper esophageal sphincter in children with type 1 laryngeal clefts after endoscopic repair via high-resolution manometry. Our study is specific as it relates to only a particular patient population. While these findings may not be able to be generalized to all children with all types of laryngeal clefts, it does suggest that the motor dynamics of the UES may be altered in children with laryngeal clefts which leads to the opportunity to develop novel therapeutic interventions in the future. More research is required to better assess whether the upper esophageal sphincter and the differences in its function may be associated with specific symptoms, comorbidities, and clinical outcomes in this particular pediatric population.

## Figures and Tables

**Figure 1 biomolecules-14-00015-f001:**
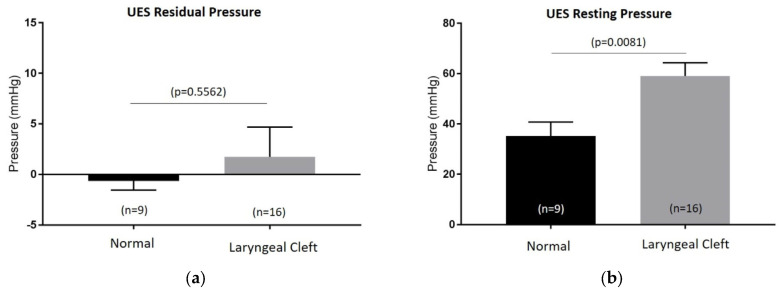
Upper-esophageal-sphincter (UES) residual (**a**) and resting pressures (**b**) measured on esophageal high-resolution manometry between control group (normal) and patients with type 1 laryngeal clefts post primary repair (laryngeal cleft).

**Figure 2 biomolecules-14-00015-f002:**
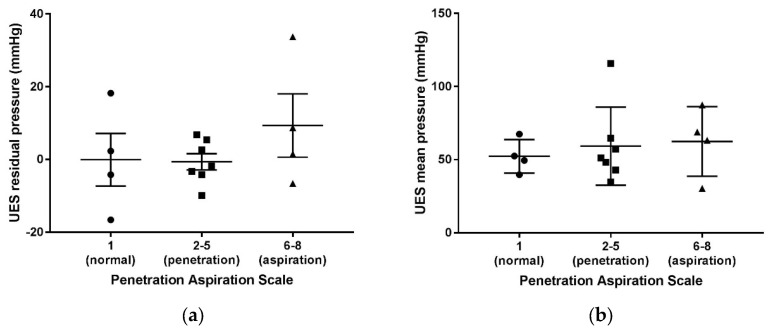
Penetration aspiration scale scores via VFSS compared to UES mean residual (**a**) and mean resting pressures (**b**) in patients with type 1 laryngeal clefts after primary repair.

**Table 1 biomolecules-14-00015-t001:** Baseline data comparison of children with type 1 laryngeal cleft post repair to age-matched controls (normal).

	Laryngeal Cleft Post Repair	Age-Matched Controls
Gender	7 Females 9 Males (*n* = 16)	4 Females 5 Males (*n* = 9)
Primary Symptom for Performing Manometry	Aspiration (68.75%) Dysphagia (18.75%) Cough (6.25%) Reflux (6.25%)	Aspiration (55%) Dysphagia (45%)
Age at Manometry (average in months old)	44.9	36.3 (*p* = 0.3784)

**Table 2 biomolecules-14-00015-t002:** Additional pharyngeal and upper-esophageal-sphincter manometric results comparing patients with type 1 laryngeal clefts post primary repair (laryngeal cleft) to age-matched control group (normal).

	Laryngeal Cleft	Normal	*p* Value
Velopharynx Maximum Pressure	135.5 ± 12.9	109.4 ± 15.1	0.2409
(mmHg with standard error of mean (SEM))	(*n* = 10)	(*n* = 5)	
Mesopharynx Maximum Pressure	142 ± 10.7	112.5 ± 14.7	0.1132
(mmHg with SEM)	(*n* = 12)	(*n* = 8)	
UES Maximum Pressure Before Relaxation	91.2 ± 7.9	66.7 ± 4.7	0.0308
(mmHg with SEM)	(*n* = 12)	(*n* = 8)	
UES Maximum Pressure After Relaxation	179.5 ± 8.9	134.9 ± 14.3	0.0118
(mmHg with SEM)	(*n* = 12)	(*n* = 8)	
UES Minimum Pressure	2.1 ± 2.0	−1.3 ± 1.2	0.2223
(mmHg with SEM)	(*n* = 12)	(*n* = 8)	
Duration of Swallow	1.03 ± 0.03	1.02 ± 0.05	0.8716
(seconds with SEM)	(*n* = 12)	(*n* = 7)	
UES Relaxation During Swallow	0.68 ± 0.04	0.79 ± 0.04	0.1014
(seconds with SEM)	(*n* = 12)	(*n* = 7)	
UES Not Relaxed During Swallow	0.35 ± 0.02	0.23 ± 0.03	0.0031
(seconds with SEM)	(*n* = 12)	(*n* = 7)	

## Data Availability

Data are contained within the article.

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
