# Peer review of "Upper Esophageal Sphincter Dysfunction in Children with Type 1 Laryngeal Cleft after Failed Primary Cleft Repair"

_biomolecules, 2023, doi:10.3390/biom14010015_

Round 1

Reviewer 1 Report

Comments and Suggestions for Authors

These authors retrospectively analyzed high-resolution esophageal manometry studies of children ≤18 years old with type 1 laryngeal clefts who had an esophageal manometry study performed for persistent symptoms after an initial primary endoscopic repair. They sought to characterize the pharyngeal and UES motor dynamics in these children. This is a novel study and adds to the literature on pediatric swallowing and esophageal function in the laryngeal cleft population. The writing is clear, organized, and concise. The background/lit review is appropriate and adequate. The methodology is appropriate and detailed. Results are clear, including tabs and figures. The discussion is thoughtful and appropriate, including limitations. I only have one question: 

1       Were any inter- intra-judge reliability measures taken for the VFSS PA scores?

Author Response

Dear Reviewer 1, 

Thank you very much for your comments and question regarding whether or not there were any inter- intra-judge reliability measures taken for the VFSS PA scores.

There were no inter- intra-reliability measures when assigning penetration-aspiration (PA) scores as there was expected to be little variability when assigning the PA scores from video fluoroscopic swallow study (VFSS) reports. PA scores were assigned by a single member of the research team using VFSS reports previously generated by our center's speech language pathologists. PA scores are rather objective as they are based off findings on the VFSS that use anatomical landmarks to describe how deep the ingested material travels down the respiratory tract in relationship to the vocal cords and whether or not the patient responds to these episodes by ejecting the material or if they do not respond. Thankfully, our center's speech language pathologists' documentation was quite clear as each report clearly identified where the ingested material traveled in the airway and in relation to the patient's vocal cords, if the material was ejected and whether or not the patient responded. Therefore, a PA score was easily able to be assigned for each VFSS and did not require an additional interpretation by the researcher to ascertain. 

Reviewer 2 Report

Comments and Suggestions for Authors

This research is the first study to characterize the upper esophageal sphincter in children with type 1 laryngeal clefts after endoscopic repair via high resolution manometry. I highly rate this study as useful for clinicians treating this disease and researchers studying the relationship between UES and aspiration. I think it would be helpful for the reader's understanding if a few more details were added.

Introduction: Please describe the benefits this research will provide to researchers and clinicians.

Line 133-140: Please describe the test diets used in VFSS.

Line 109: Which company's product is "ManoScan 360 High-resolution Manometry System software-ware"?

Line 169: Is ESMO an abbreviation? Please enter this official name.

Author Response

Dear reviewer 2, 

Thank you very much for your comments and questions. Please see our responses to your questions and suggestions below:

-"Introduction: Please describe the benefits this research will provide to researchers and clinicians."

Response: Thank you for pointing this out as we do believe that this research is important for its potential clinical implications. Please see the edits in regards to this suggestion added in the Introduction section of the revised manuscript on lines 91-97.

-"Line 133-140: Please describe the test diets used in VFSS."

Response: Test diets provided during each individual VFSS followed our institutions standard protocol as it was at the discretion of the speech language pathologist performing the procedure. All patients were tested with thin and/or nectar thick liquids. Patients who were previously thickening feeds of nectar thick consistency (most often due to a previous safety concern with thin liquids) prior to the VFSS performed, were maintained on this. Although these were the minority of this group, (3 out of 14 patients). Please see updated text on lines 144-145 in the edited manuscript which respond to this comment. 

-"Line 109: Which company's product is "ManoScan 360 High-resolution Manometry System software-ware"?"

Response: Thank you for identifying this. The software used was from Medtronic. Please see that this is now included in the edited manuscript which is now on line 114 due to other edits that have been made. 

-"Line 169: Is ESMO an abbreviation? Please enter this official name."

Response: Thank you for identifying this. ESMO is an abbreviation for esophageal manometry. Please see this edit in the edited manuscript. Due to other changes made to the manuscript, this edit can now be seen on line 174 in the updated manuscript. 

Thank you again for your comments and suggestions.